# Series of Organotin(IV) Compounds with Different Dithiocarbamate Ligands Induced Cytotoxicity, Apoptosis and Cell Cycle Arrest on Jurkat E6.1, T Acute Lymphoblastic Leukemia Cells

**DOI:** 10.3390/molecules28083376

**Published:** 2023-04-11

**Authors:** Nur Rasyiqin Rasli, Asmah Hamid, Normah Awang, Nurul Farahana Kamaludin

**Affiliations:** 1Program of Biomedical Science, Center for Toxicology and Health Risk Studies, Faculty of Health Sciences, Universiti Kebangsaan Malaysia, Jalan Raja Muda Abdul Aziz, Kuala Lumpur 50300, Malaysia; p105535@siswa.ukm.edu.my; 2Program of Environmental Health and Industrial Safety, Center for Toxicology and Health Risk Studies, Faculty of Health Sciences, Universiti Kebangsaan Malaysia, Jalan Raja Muda Abdul Aziz, Kuala Lumpur 50300, Malaysia; norm@ukm.edu.my (N.A.); nurulfarahana@ukm.edu.my (N.F.K.)

**Keywords:** childhood leukemia, organotin(IV), dithiocarbamate, cytotoxicity, cell death, cell cycle arrest

## Abstract

The discovery of cisplatin has influenced scientists to study the anticancer properties of other metal complexes. Organotin(IV) dithiocarbamate compounds are gaining attention as anticancer agents due to their potent cytotoxic properties on cancer cells. In this study, a series of organotin compounds were assessed for their toxic effects on the Jurkat E6.1 cell line. WST-1 assay was used to determine the cytotoxic effect of the compounds and showed that six out of seven organotin(IV) dithiocarbamate compounds exhibited potent cytotoxic effects toward T-lymphoblastic leukemia cells, Jurkat E6.1 with the concentration of IC_50_ ranging from 0.67–0.94 µM. The apoptosis assay by Annexin V-FITC/PI staining showed that all tested compounds induced cell death mainly via apoptosis. Cell cycle analysis assessed using RNase/PI staining showed that organotin(IV) dithiocarbamate compounds induced cell cycle arrest at different phases. In conclusion, the tested organotin(IV) dithiocarbamate compounds demonstrated potent cytotoxicity against Jurkat E6.1 cells via apoptosis and cell cycle arrest at low IC_50_ value. However, further studies on the mechanisms of action are required to probe the possible potential of these compounds on leukemia cells before they can be developed into anti-leukemic agents.

## 1. Introduction

Leukemia is known as bone marrow and blood cancer. Four common types of leukemia are acute lymphoblastic leukemia (ALL), chronic lymphoblastic leukemia (CLL), acute myeloid leukemia (AML), and chronic myeloid leukemia (CML). ALL with a prevalence of 25% of cancers in children under the age of 15 years, is the most common cancer in childhood. This ALL has a 93.5% five-year survival rate when treated with the newest protocol and top chemotherapy [1]. ALL occurs due to the malignancy of B or T lymphoblasts that are characterized by uncontrolled proliferation of abnormal, immature lymphocytes and their progenitors [2]. Pediatrics with ALL originating from B cells are diagnosed in up to 85% of cases, while cases forming from T cells comprises 15% [3].

Clinically, various anticancer drugs are used to treat ALL patients [4]. However, some treatments given can cause adverse effects. Vincristine, for example, is widely used to treat childhood leukemia. However, this drug was reported to cause toxicity to nervous system (neuropathy), and adverse reaction towards gastrointestinal and bone marrow suppression [5]. Moreover, there are many cases of resistance toward drug therapy detected in patients with ALL [6].

The toxicity research on metal-based substances become a hot topic after the discovery of cisplatin as a potent chemotherapy drug towards various cancer cells [7,8]. Organotin compounds are tin (Sn)-based compounds, and have been widely used in industry as components of pesticides, fungicides, bactericides, and as plastic stabilizers [9]. Numerous research has also been conducted to investigate their biological activities, including antitumor properties and mechanisms of action against solid and non-solid cancer cells as part of the development process of new anticancer drugs [10,11,12]. Moreover, different toxicities are displayed depending on the number and nature of functional groups attached to the tin atom [13].

The common types of cell death that occur in the body are apoptosis and necrosis [14]. Apoptosis is a normal physiological process of cell death [15]. This type of cell death is responsible for tissue remodeling during the development and turnover of normal tissue (e.g., hematopoietic cells) throughout the life span of multicellular organisms [16]. The mechanisms and characteristics of apoptosis include chromatin condensation, nuclear fragmentation, membrane blebbing, and cell shrinkage [17]. As for necrosis, the observable characteristics are cell swelling, nuclear membrane disruption, inflammation of the cells, and lysis of the nuclear chromatin [18]. According to Yamaguchi and colleagues [19], a compound that can induce cell death via apoptosis has the potential to be developed into anticancer drugs. Any compound that arrests the cell cycle progression is also considered a promising drug mechanism to combat cancer cells [20].

Therefore, seven organotin(IV) dithiocarbamate compounds, which are diphenyltin(IV) diisopropyl dithiocarbamate (ODTC 1), diphenyltin(IV) diallyl dithiocarbamate (ODTC 2), triphenyltin(IV) diisopropyl dithiocarbamate (ODTC 3), triphenyltin(IV) diallyl dithiocarbamate (ODTC 4), triphenyltin(IV) diethyl dithiocarbamate (ODTC 5), dimethyltin(IV) diisopropyl dithiocarbamate (ODTC 6) and dimethyltin(IV) diethyl dithiocarbamate (ODTC 7) (Figure 1) were investigated for their cytotoxic effects against Jurkat E6.1 human T acute lymphoblastic leukemia cell line, to evaluate their potential as future therapeutic anticancer agents.

## 2. Results

### 2.1. Cytotoxic Effects of Organotin(IV) Dithiocarbamate Compounds and Vincristine (Positive Control) on Jurkat E6.1 Cell Line

The percentage of viability for cells treated with organotin(IV) dithiocarbamate is shown in Figure 2. The graph showed that, organotin(IV) dithiocarbamate compounds could reduce 50% of Jurkat E6.1 cell viability at a concentration below 4.0 µM except for ODTC 6. As for WIL2-NS cells, the organotin(IV) dithiocarbamate compounds were able to reduce half of the cell population below 5.0 µM.

The IC_50_ values obtained from organotin(IV) dithiocarbamate compounds and Vincristine on Jurkat E6.1 and WIL2-NS cell line are summarized in Table 1. Results from WST-1 assay showed that all of the triphenyltin(IV) compounds, displayed the most toxic effect towards Jurkat E6.1 cells as shown by the IC_50_ value below 1.0 µM (0.67–0.94 µM). The diphenyltin(IV) dithiocarbamate compounds produced a less toxic effect on the cells with IC_50_ values ranging from 1.05–1.45 µM. The dimethyltin(IV) dithiocarbamate compounds produced the least toxic effect for this series of organotin(IV) compounds toward Jurkat E6.1 cells. The cytotoxic graph of vincristine towards Jurkat E6.1 cells showed a plateau patent after the viability cell reach 40% at 0.30 µM concentration up to the highest concentration (5.00 µM). This clearly indicates that vincristine caused a cytostatic effect on the cell not cytotoxic like organotin(IV) dithiocarbamate compounds. Therefore, the organotin used is much better than vincristine in terms of cytotoxic effects. Cells will slowly proliferate or grow back once the cytostatic effects have faded away.

The selectivity index (SI) showed that ODTC 3 has the highest value (SI = 1.11) followed by ODTC 1 and vincristine (SI = 0.96). ODTC 5 and 4 produced SI value of 0.86 and 0.71 respectively. The SI value obtained by ODTC 2 is 0.50. And finally, ODTC 7 has the lowest SI with a value of 0.15.

### 2.2. Mode of Cell Death of Jurkat E6.1 Cells Treated with Organotin(IV) Dithiocarbamate Compounds and Vincristine

The mode of cell death of Jurkat E6.1 cells treated with organotin(IV) dithiocarbamate is shown in Figure 3. In general, the organotin(IV) dithiocarbamate compounds cause significant cell death at the apoptotic phase, with very few necrotic populations.

From the results obtained, the triphenyltin(IV) series caused majority of cell death at apoptosis phase. ODTC 3 displayed a total of 87.5% apoptosis with 59.70 ± 1.96% of cell population in early apoptotic phase and 27.80 ± 2.17% in late apoptotic phase. ODTC 4 showed that 49.70 ± 2.84% of cells are in the early apoptosis stage while 38.63 ± 2.73% in late apoptosis. To sum up, the total apoptosis population caused by ODTC 4 on Jurkat E6.1 cells is 88.33%. As for ODTC 5, the total apoptosis population on Jurkat E6.1 cells is 82.97 ± 2.84%, with 56.50 ± 1.64% of it in early apoptosis and 26.47 ± 1.23% in late apoptosis. For the diphenyltin(IV) and dimethyltin(IV) series, the majority of the cell population detected is viable (52.23–73.97%). ODTC 1 only caused 25.4% of cell death at the apoptotic phase with 12.23 ± 1.93% of the cell population in early apoptosis and 13.17 ± 1.13% in late apoptosis. For ODTC 2, the cells were in the early apoptotic phase with 17.20 ± 2.69% and 30.07 ± 2.38% late apoptotic, with a total of the apoptotic population is 47.27%. ODTC 7 causes 28.06% apoptosis with 13.13 ± 2.62% in early apoptosis and 14.93 ± 1.35% in late apoptosis. ODTC 6 does not been tested for its mode of cell death since this compound only caused a cytostatic effect and does not produce IC_50_ value at the tested range of concentration.

Treatment of vincristine (positive control) towards Jurkat E6.1 cells revealed that the population of cells treated also died at the apoptotic phase (54.64%). Vincristine induced early apoptosis (36.07 ± 2.85%) of the cells while 18.57 ± 1.31% in the late apoptosis population. 

### 2.3. Cell Cycle Analysis of Jurkat E6.1 Cells Treated with Organotin(IV) Dithiocarbamate Compounds and Vincristine

To investigate the cell cycle progression of Jurkat E6.1 cells treated with organotin(IV) dithiocarbamate compounds, evaluation by flow cytometry was conducted using the RNase/PI stain for 24 h of exposure. Different treatments cause different effects on Jurkat E6.1 cells as shown in Figure 4 and the summary of organotin(IV) dithiocarbamate action on the cell cycle phases is shown in Table 2. Based on the three independent experiments, it was found that the treatment of ODTC 1 on Jurkat E6.1 does not cause cell cycle arrest at any phase. Treatment of ODTC 2, ODTC 4, ODTC 5 and ODTC 7 caused cell cycle arrest at the S-G2/M phase. Only ODTC 3 and vincristine caused cell cycle arrest at G0/G1 phase. ODTC 6 does not been tested for its cell cycle activity since this compound does not produce IC_50_ value at the tested range of concentration.

The results showed that 59.37%  ± 3.07 Jurkat E6.1 cells that were exposed to ODTC 2 were arrested in the S phase while 19.03%  ±  0.20 and 21.63%  ±  3.17 of treated cells were distributed in G2/M and G0/G1 phase respectively. Treatment with ODTC 4 displayed that 64.10%  ± 4.46 of Jurkat E6.1 cells were accumulated in the S phase while cells in G0/G1 and G2/M phase were 13.22%  ±  2.30 and 22.67% ±  3.54 respectively. For treatment with ODTC 5, a number of Jurkat E6.1 cells accumulated in the S phase (58.70% ± 3.37). Only 9.84% ± 0.71 of the cell population accumulated in the G0/G1 phase and 31.46% ± 2.86 in the G2/M phase. As for ODTC 7, the treatment caused the arrestment of the cell cycle at the S phase with 57.19% ± 2.89. Cells distributed in G2/M and G0/G1 phase are 17.39% ± 4.52 and 23.41% ± 5.37 respectively. Other than that, 68.37% ± 4.78 of the Jurkat E6.1 cells treated with ODTC 3 were arrested at G0/G1 phase. In addition, a small percentage of the cells (27.29% ± 5.12 and 4.34% ± 0.99) were in S and G2/M phase. Jurkat E6.1 cells treated with vincristine (the positive control) also exhibited significant changes when compared to the negative control, specifically at the G0/G1 and S checkpoints at 24 h. The population of cells treated with vincristine detected in G0/G1 is 66.76% ± 1.39 with 25.24% ± 1.39 in S phase and 8.00% ± 0.00 in the G2/M phase.

## 3. Discussion

This study was performed to assess the cytotoxic effects of organotin(IV) dithiocarbamate compounds which are diphenyltin(IV) diisopropyl dithiocarbamate (ODTC 1), diphenyltin (IV) diallyl dithiocarbamate (ODTC 2), triphenyltin(IV) diisopropyl dithiocarbamate (ODTC 3), triphenyltin(IV) diallyl dithiocarbamate (ODTC 4), triphenyltin(IV) diethyl dithiocarbamate (ODTC 5), dimethyltin(IV) diisopropyl dithiocarbamate (ODTC 6) and dimethyltin(IV) diethyl dithiocarbamate (ODTC 7) and also vincristine (positive control) against Jurkat E6.1 human acute T lymphoblastic leukemia cell line using WST-1 assay (Figure 2). The most potent organotin(IV) dithiocarbamate compound in this study is ODTC 3. The cytotoxic effect was also measured on non-cancerous cells, WIL2-NS.

The cytotoxic effect and reaction of the organotin(IV) dithiocarbamate compounds are closely related to their structure and properties. The chemical structure of triphenyltin(IV) dithiocarbamate consists of three phenyl groups attached to the tin (Sn) atom. This makes the compound more lipophilic and exerts a more potent effect on the biological system compared to phenyltin or diphenyltin [23]. The number of the phenyl group in the parent structure explained the toxic effect of the organotin(IV) compounds and also characterized as electrophiles [24]. Thus, the toxicity of these compounds is believed to be due to their interaction with electron donor groups in biomolecules. Besides the presence of the phenyl group, the metal-ligand interaction also influences the reaction of the organotin(IV) complex toward cell biology [25]. According to the World Health Organization [26], trisubstituted alkyl and aryl(IV) compounds are more toxic than di-substituted organotin(IV) compounds. In contrast, monosubstituted organotin(IV) compounds are the least harmful organotin(IV). This explains the high magnitude of toxicity of diphenyltin(IV) and triphenyltin(IV) toward Jurkat E6.1 cells.

The longer the hydrocarbon chain, the more potent the compound [27]. In this study, ODTC 3 is the most potent in the triphenyltin(IV) series because the diisopropyl structure contains 6 carbon atoms and a single bond whereas the diethyl structure of ODTC 5 has four carbon atoms, and the diallyl structure of ODTC 4 contains a double bond. Allylic C−H bonds are weaker and more reactive than those in ordinary sp3 carbon centres. This structure also influences the cytotoxic effect and the compounds’ potency level.

Generally, dithiocarbamate compounds have a lipophilic character that makes it easier for them to interact with cells and to cross the biological systems’ membranes [28]. Also, the lipophilicity environment influences the receptor-binding complex and complex formed between the compounds [29]. Spreckelmeyer et al. [30] suggested that the lipophilic character of the compound has aided in the intracellular interaction and caused cytotoxicity towards the cells.

A similar study by Kamaludin et al. [31] also showed that the organotin(IV) N-butyl-N-phenyldithiocarbamate compounds and the positive control used (doxorubicin) produced a comparable effect on Jurkat E6.1 cells. The value of IC_50_ produced by the organotin(IV) compounds in their study ranged between 0.4–1.3 μM while doxorubicin caused a cytotoxic effect at 0.1 μM. Another study done by Kamaludin et al. [11] showed that organotin(IV) (2-metoxyethyl) methyldithiocarbamate compound was able to produce a potent cytotoxic effect on several human leukemia cell lines. As for the HL-60 (acute promyelocytic leukemia) cell line, the IC_50_ values of compounds that cause cytotoxic effects are in the range of 0.06–0.18 µM, while for the Jurkat E6.1, the range was 0.14–1.30 µM. Meanwhile, the compound produced an IC_50_ value in the range of 5.20–5.40 µM against K562 (myelomonocytic leukemia cells) because this cell is known as apoptosis-resistant cells. Awang et al. [32] carried out a study on the cytotoxic activity of newly synthesized triphenyltin(IV) alkylisopropyl di thiocarbamate compounds on Jurkat E6.1 human acute T-lymphoblastic cell line, and found that the IC_50_ values are ranged between 0.03–0.41 µM for exposure times of 24, 48, and 72 h. This showed that the synthesized organotin(IV) dithiocarbamate compounds could induce a cytotoxic effect on different leukemia cell lines even at low doses.

The treatment of vincristine on Jurkat E6.1 cells caused cytostatic effects up to the highest concentration. Cytostatic effect means, a substance can slow down or arrest the growth of the cells without killing them [33]. Once the treatment stops, the cells will be able to grow back and spread. As for this study, the cytotoxic effects of organotin(IV) dithiocarbamate is better than vincristine because the compounds used are able to kill the Jurkat E6.1 cells and not just arrest the cell growth which might cause the cells to spread after the treatment is stopped.

Selectivity index (SI) is the reference value indicating the selectivity of the compound/substances towards cancerous cells over non-cancerous cells. According to Bartmańska et al. [34] the SI value that is higher than 1.00 implies that the compound is selective towards cancerous cells and considerate as anticancer-specific. From the SI calculation, only ODTC 3 displayed selective activity towards Jurkat E6.1 cells (SI = 1.11). Other organotin(IV) dithiocarbamate compounds treated on Jurkat E6.1 cells produced SI of less than 1.00. This showed that ODTC 3 has good potential to kill leukemia cells and causes less negative effect on non-cancerous cells. However, the cell death mechanism of action and its pathway need to be explored more in order to understand deeper of its bioavailability and pharmacodynamics.

Cell death can be classified into a few different forms which are apoptosis, autophagy, necrosis, and entosis [35]. However, in this study, our focus on cell death mechanism is only on apoptosis and necrosis. Apoptosis is a type of cell death triggered by physical, chemical, and biological factors [36]. The cellular response to this type of cell death is tightly regulated [37]. Apoptosis plays an essential role in many pathological conditions, particularly cancer, and neurodegeneration [38]. Necrosis is an irreversible and uncontrolled cell death that caused swelling of the cell organelles, plasma membrane rupture, and spillage of intracellular contents into the surrounding area leading to tissue damage [18]. Flow cytometry is considered an analytical tool for investigating and evaluating the potency of a drug or substances, in addition to determining the cell viability, membrane and chromosomal damage evaluation, cell-cycle analysis, and morphological changes [12]. Hence, as an extension of the WST-1 assay, a quantification analysis was conducted to evaluate the apoptotic and necrotic cell death population using a flow cytometer.

From the mode of cell death assessment, it was found that the organotin(IV) dithiocarbamate compounds caused the Jurkat E6.1 cell death via apoptosis. This result is quite similar to many other studies where the organotin(IV) compound induced cell death via apoptosis [39] where it shows that an organotin indomethacin derivative significantly inhibited cell proliferation and activated apoptosis by regulating the MAPK, activating caspase 3, and the cytokine immune mediator, IL-6, in breast and prostate cancer cells. Other than that, Haezam et al. [22] reported that two organotin(IV) diallyl dithiocarbamate compounds showed potent cytotoxicity in HT-29 (colon adenocarcinoma) cells, and can kill the cancer cell line through apoptosis. Both compounds also have good selectivity towards the HT-29 cells while not the CCD-18Co (non-cancerous colon) cell line. A different study done by Kadu et al. [12] showed that the diphenyltin(IV) dithiocarbamate compound synthesized, was able to induce 0.7–95% of pro-apoptosis (early apoptosis) towards IMR 32 (neuroblast) cells and 4.3–45.7% of pro-apoptosis towards HEP 3B (hepatocellular carcinoma) cells. As for the apoptosis (late apoptosis) population, these compounds can induce 3.8–65% of the IMR 32 cell population and 29.3–66.9% of the HEP 3B cell population.

In normal and healthy cells, there’s a structure called phosphatidylserine (PS) that is located in the inner membrane of the phospholipid bilayer. However, when apoptosis occurs, PS will be flipped and translocated to the outer leaflet of the plasma membrane, exposed it to the external cellular environment [40]. It is suggested that, the treatment of organotin(IV) dithiocarbamate compound causes the loss of plasma membrane integrity and the PS to be detected, thus can bind to the Annexin V FITC fluorochromes stain.

Originally, apoptosis is defined by the morphology and characteristic of dying cells [41]. However, the morphology of apoptosis and necrosis is distinctively different. Shrinkage of the cells, chromatin condensation, and small fragmentation of disintegrating cells are signs of a cell leading to apoptotic death [42]. In necrosis, cell swelling is caused by the excessive fluid influx, while shrinkage of the cells can be observed in the cells that undergo apoptosis [43]. Besides that, during apoptosis, chromatin aggregation and condensation can be observed around the nuclear membrane but not for necrotic cells [41]. The membrane of the cell remains intact and shows an irregular budding structure known as membrane blebbing during apoptosis. Meanwhile, in necrotic cells, the intracellular content released from the cells causes them to lose integrity and eventually damage the cell membrane [44].

The cell cycle progression of Jurkat E6.1 cells treated with organotin(IV) dithiocarbamate compounds was evaluated using flow cytometry after reaction with the RNase/PI stain for 24 h. ODTC 3 and vincristine caused cell cycle arrest at G0/G1 checkpoint, a phase where the cell is growing and proteins and RNA are synthesized. Arrested cells at this phase cause the cell to not enter the S phase and this eventually stops the cells from duplicating. As for ODTC 2, ODTC 4, ODTC 5 and ODTC 7, the cell cycle arrest at the S-G2/M phase. S phase in the cell cycle is a phase where the genetic material is synthesized and replicated. G2/M checkpoint is a phase where it has the function of preventing cells with damaged DNA, lasting from the G1 and S phases or generated in G2, from entering mitosis phase. Arrested cells at these phases cause the cell unable to produce two new daughter cells. In this study, the treatment of ODTC 1 towards Jurkat E6.1 cells does not cause cell cycle arrest at any phase. This showed that ODTC 1 is not a cell cycle specific.

There are many research works conducted to study and investigate the effects of different organotin(IV) dithiocarbamate compounds on the cell cycle activities of different cell lines. For example, a cyclic trinuclear organotin(IV) complex with an aromatic oximehydroxamic acid group was able to cause cell cycle delay in the late S phase and G2/M checkpoint of HCT116 cells [45]. Meanwhile, a diorganotin(IV) complex with a 4-nitro-N-phthaloyl-glycine group induced the cell cycle arrest of HepG-2 cells at the G2/M phase [46]. So, we can conclude from this assay that different compound causes different effects and different target in the cell cycle phase.

## 4. Materials and Methods

### 4.1. Chemicals and Materials

Rosewell Park Memorial Institute (RPMI) 1640 media (Nacalai Tesque, Kyoto, Japan), fetal bovine serum (FBS) (Tico Europe, Amstelveen, The Netherlands), 1% penicillin-streptomycin (Nacalai Tesque, Kyoto, Japan), absolute ethanol (Systerm, Selangor, Malaysia), 2-(4-iodophenyl)-3-(4-nitrophenyl)-5-(2,4-disulfophenyl)-2H-tetrazolium monosodium salt (WST-1) (Roche, Basel, Switzerland), dimethylsulfoxide solution (DMSO) (Sigma, St. Louis, MO, USA), trypan blue 0.4% (Sigma, St. Louis, MO, USA), phosphate buffer solution (PBS) (Medicago, Uppsala, Sweden), Vincristin (ChemFaces, Hubei, China), Annexin V-FITC/PI kit (BD Bioscience, East Rutherford, NJ, USA), RNase/PI stain (BD Bioscience, East Rutherford, NJ, USA), L-glutamate (Sigma, St. Louis, MO, USA), organotin(IV) dithiocarbamate compounds.

### 4.2. Compounds

Seven organotin(IV) dithiocarbamate compounds, which are diphenyltin(IV) diisopropyl dithiocarbamate (ODTC 1), diphenyltin(IV) diallyl dithiocarbamate (ODTC 2), triphenyltin(IV) diisopropyl dithiocarbamate (ODTC 3), triphenyltin(IV) diallyl dithiocarbamate (ODTC 4), triphenyltin(IV) diethyl dithiocarbamate (ODTC 5), dimethyltin(IV) diisopropyl dithiocarbamate (ODTC 6) and dimethyltin(IV) diethyl dithiocarbamate (ODTC 7) were synthesized and characterized by our team (Haezam et al.) [22,47,48] at the School of Chemical Sciences and Food Technology, Faculty of Science and Technology, Universiti Kebangsaan Malaysia, Bangi, Malaysia.

### 4.3. Organotin Compound Stock Preparation

All stock solutions of the organotin compounds were prepared at 30 mM and dissolved in dimethyl sulphoxide (DMSO). The stock solutions were stored at 4 °C, fresh dilutions were made by adding the medium before the experiment.

### 4.4. Cell Culture

The Jurkat E6.1 (TIB-152™) and WIL2-NS (CRL-8155™) cell lines were purchased from the American Type Culture Collection. These cells were cultured in Roswell Park Memorial Institute Medium 1640, RPMI-1640 supplemented with 10% Fetal Bovine Serum (FBS) and 1% penicillin-streptomycin to obtain the complete growth medium. The cells were incubated at 37 °C with 5% CO_2_ (New Brunswick Galaxy 170 S, Hamburg, Germany).

### 4.5. WST-1 [2-(4-Iodophenyl)-3-(4-nitrophenyl)-5-(2,4-disulfophenyl)-2H tetrazolium] Assay 

Jurkat E6.1 and WIL2-NS cell lines were cultured in 75 cm^2^ flasks and seeded in 96-well culture plates with a final concentration of 1 × 10^6^ cells/mL. The cells were exposed to organotin(IV) dithiocarbamate compounds with the highest concentration of 5.0 μM. Vincristine was used in this study as a positive control with its highest concentration, at 5.0 μM, while untreated cells (cells and complete medium only) were used as the negative control. Serial dilution methods are used to obtain the desired concentration gradient and the cells were then exposed to the compounds overnight (24 h). After the incubation period, the WST-1 solution (Roche, Basel, Switzerland) was added in a ratio of 1:10 to the suspension volume in the wells and incubated for another 4 h. Finally, the cells’ optical density (OD) was measured using an ELISA microplate reader (Multiskan FC, Thermo Scientific, Waltham, MA, USA) at 440 nm wavelength. The data obtained were analyzed using GraphPad Prism software (Dotmatics) to determine the IC_50_ values.

### 4.6. Selectivity Index (SI)

The selectivity index is determined by dividing the cytotoxic activity (IC_50_ value) of each compound on a non-cancerous cell line (WIL2-NS) to a leukemia cell line (Jurkat E6.1) using the following equation:SI = IC_50_ for non cancerous cell line
IC_50_ for cancerous cell line

### 4.7. Annexin V FIT-C/PI Stain Assay

Apoptosis assay was performed according to the manufacturer’s instructions using Annexin V-FITC/PI staining kit (BD Biosciences, East Rutherford, NJ, USA). The Jurkat E6.1 cells were seeded in a 6-well plate with a final concentration of 5.0 × 10^5^ cell/mL. The cells were then treated with organotin(IV) dithiocarbamate compounds and vincristine at their respective IC_50_ concentrations and then incubated for 24 h. After the incubation period, 500 μL of cells were harvested and transferred into a microcentrifuge tube. The sample was centrifuged at 1500 rpm for 5 min at 4 °C (Mikro 200R, Hettich, Kirchlengern, Germany) and the supernatant was discarded, and 500 μL of cold PBS was added to the sample followed by another centrifugation for 5 min, then the supernatant was discarded. Next, 150 μL of Annexin V binding buffer and 5.0 μL of Annexin V-FITC were added to the sample in the dark and then incubated for 15 min at room temperature. After that, 3.0 μL of propidium iodide (PI) was added to the samples. The samples then incubated for another 2 min at room temperature. Finally, 350 μL of Annexin V binding buffer was added to the samples and transferred into Falcon tubes. The samples were then analyzed using a BD FACS Canto II flow cytometer (BD Bioscience, East Rutherford, NJ, USA).

### 4.8. Cell Cycle Analysis

Cell cycle analysis was performed according to the manufacturer’s instructions using PI/RNase Staining Buffer (BD Bioscience, East Rutherford, NJ, USA). For this assay, the Jurkat E6.1 cell line was cultured in 6-well culture plates with a final concentration of 1.0 × 10^6^ cells/mL and treated with organotin(IV) dithiocarbamate compounds for 24 h, with vincristine as the positive control. The concentration of each treatment used was based on the IC_50_ values obtained from the WST-1 assay. After the incubation period, 500 µL of cell suspension was washed twice with cold PBS and fixed overnight with 500 µL of 70% ethanol. The cells were then stained with 500 µL RNase/PI stain and measured using a BD FACS Canto II flow cytometer (BD Bioscience, East Rutherford, NJ, USA). The cell cycle phases were analyzed using the ModFit LT software (Verity Software House).

### 4.9. Statistical Analysis

All data are presented as the mean ± standard error of mean (SEM). Statistical analysis and evaluation of the percentage of viable cells and the concentration of compounds used to treat the cells were calculated using GraphPad Prism 9 software (Dotmatics) by employing a one-way analysis of variance (ANOVA). A *p*-value of < 0.05 is considered statistically significant.

## 5. Conclusions

This study has proven that the organotin(IV) dithiocarbamate compounds exhibited a cytotoxic characteristics because of their ability to inhibit Jurkat E6.1 cell growth at low doses with the most potent compound being ODTC 3. The compounds were also able to induce cell death via apoptosis after 24 h of treatment with the IC_50_ concentration. Further assessment revealed that the compounds demonstrated cell cycle arrest at different phases. Thus, this study indicates that the organotin(IV) dithiocarbamate compounds possess good potential to be developed into new anti-leukemic agents.

## Figures and Tables

**Figure 1 molecules-28-03376-f001:**
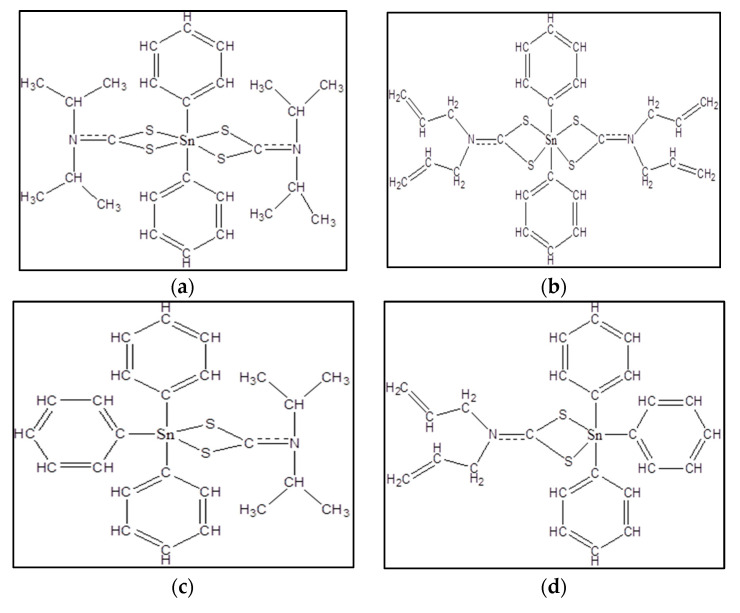
Chemical structure of (**a**) ODTC 1, (**b**) ODTC 2, (**c**) ODTC 3, (**d**) ODTC 4, (**e**) ODTC 5, (**f**) ODTC 6, and (**g**) ODTC 7 [21,22].

**Figure 2 molecules-28-03376-f002:**
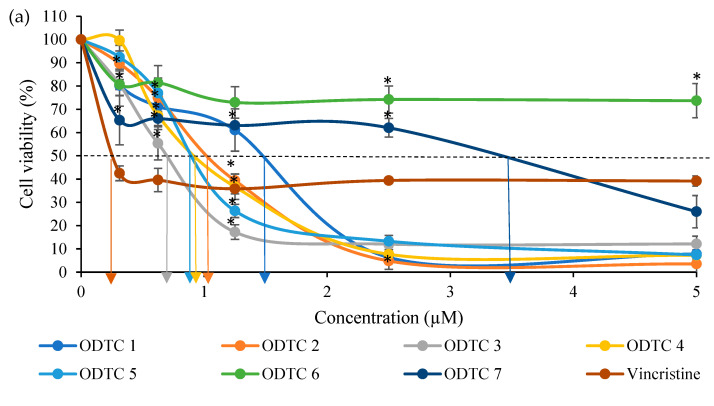
Cytotoxic effects of diphenyltin(IV) diisopropyl dithiocarbamate (ODTC 1), diphenyltin(IV) diallyl dithiocarbamate (ODTC 2), triphenyltin(IV) diisopropyl dithiocarbamate (ODTC 3), triphenyltin(IV) diallyl dithiocarbamate (ODTC 4), triphenyltin(IV) diethyl dithiocarbamate (ODTC 5), dimethyltin(IV) diisopropyl dithiocarbamate (ODTC 6) and dimethyltin(IV) diethyl dithiocarbamate (ODTC 7) towards (**a**) Jurkat E6.1 and (**b**) WIL2-NS cell line after 24 h of exposure. The IC_50_ values obtained are as shown in the graph above. The data presented are the mean percentage of viable cells (%) ± SEM obtained from three different experiments (*n* = 3). * Significant difference (*p* < 0.05) from the negative control.

**Figure 3 molecules-28-03376-f003:**
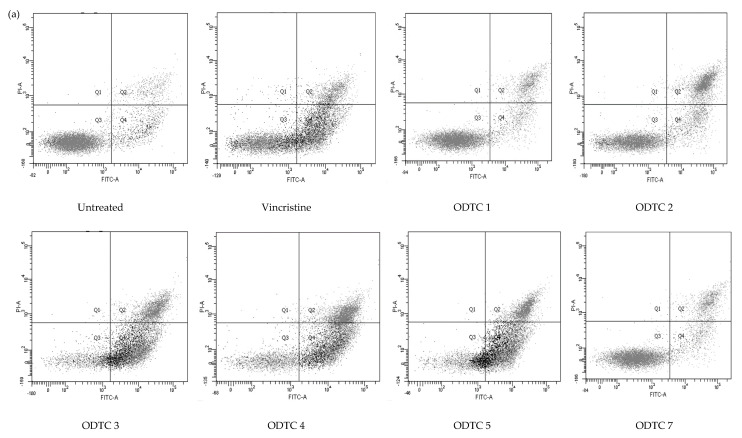
(**a**) Flow cytometry analysis of Jurkat E6.1 cells stained with Annexin V−FITC/PI after treatment with ODTC 1, ODTC 2, ODTC 3, ODTC 4, ODTC 5, ODTC 7 and vincristine at their respective IC_50_ concentrations. (**b**) The percentage of viable, apoptotic (early and late), and necrotic populations are as depicted. The data show the mean percentage of cells (%) ± SEM obtained from three independent experiments. * Significant difference (*p* < 0.05) from the negative control.

**Figure 4 molecules-28-03376-f004:**
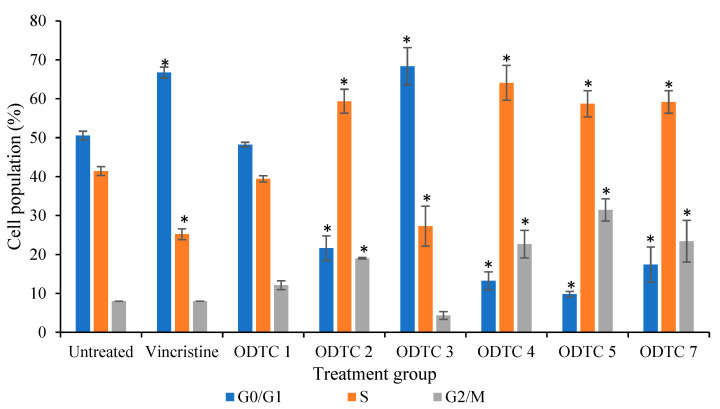
Flow cytometry analysis of Jurkat E6.1 cells stained with RNase/PI and treated with ODTC 1, ODTC 2, ODTC 3, ODTC 4, ODTC 5, ODTC 7 and vincristine at their respective IC_50_ concentrations for 24 h. Percentage (%) of cells in G0/G1, S and G2/M is depicted. Data values are the mean ± SEM percentage from three independent experiments. * Significant difference (*p* < 0.05) from the negative control.

**Table 1 molecules-28-03376-t001:** The IC_50_ values of organotin(IV) dithiocarbamate compounds and vincristine against Jurkat E6.1 cells.

Treatment	Value of IC_50_ (μM)		Selectivity Index (SI)
	Jurkat E6.1	WIL2-NS	
Vincristine (positive control)	0.24 ± 0.02	0.23 ± 0.03	0.96
ODTC 1	1.45 ± 0.02	1.39 ± 0.46	0.96
ODTC 2	1.05 ± 0.02	0.52 ± 0.14	0.50
ODTC 3	0.67 ± 0.06	0.75 ± 0.07	1.11
ODTC 4	0.94 ± 0.08	0.67 ± 0.14	0.71
ODTC 5	0.92 ± 0.05	0.79 ± 0.05	0.86
ODTC 6	NA	5.0 ± 0.02	NA
ODTC 7	3.40 ± 0.25	0.50 ± 0.02	0.15

**Table 2 molecules-28-03376-t002:** Summary of organotin(IV) dithiocarbamate action on the cell cycle phases.

Treatment	Phases of Cell Cycle Arrest
Untreated	-
Vincristine (positive control)	G0/G1
ODTC 1	Not cell cycle specific
ODTC 2	S-G2/M
ODTC 3	G0/G1
ODTC 4	S-G2/M
ODTC 5	S-G2/M
ODTC 6	-
ODTC 7	S-G2/M

## Data Availability

All data used to support the findings of this study are included in the article.

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
