# Peer review of "Series of Organotin(IV) Compounds with Different Dithiocarbamate Ligands Induced Cytotoxicity, Apoptosis and Cell Cycle Arrest on Jurkat E6.1, T Acute Lymphoblastic Leukemia Cells"

_molecules, 2023, doi:10.3390/molecules28083376_

Round 1
Reviewer 1 Report
I reviewed the manuscript "Series of organotin(IV) compounds with different dithiocarbamate ligands induced cytotoxicity, apoptosis and cell cycle arrest on Jurkat E6.1, T acute lymphoblastic leukemia cells" written by Nur Rasyiqin Rasli et al. In my opinion the manuscript is interesting but still needs minor changes to be accepted for publication in Molecules. My main observation on the manuscript is the lack of necessary citation for several important sentences and the request to improve its language by removing unnecessary (repeated) words. Here I give examples on sentences with unnecessary words or missing citation, but the authors need to carefully check the whole manuscript for all these sentences.
Line 11: The discovery of cisplatin ha influenced the scientist to study the anticancer properties of other metal complexes should be "The discovery of cisplatin ha(s) influenced the scientist(s) to study the anticancer properties of other metal complexes"
Line 33: This ALL ha 93.5% of five-year survival rate when treated with should change to "This ALL ha(s) 93.5% of five-year survival rate when treated with".
Line 38: Clinically, various anticancer treatment are used to kill the cancerous cells [4]. Please remove "to kill the cancerous cells" as this is clear & treatment should change to "treatments".
The sentences "The common types of cell death that occur in the body are apoptosis and necrosis." Line 53 and "Apoptosis is a normal physiological process of cell death." Line 54 and many other sentences miss citation.
In the Chemicals and materials section (line 91), the authors provided organotin(IV) dithiocarbamate compounds, but they never mentioned their source/ did not describe the preparation procedures.
The methods followed/programs used in this research need to be cited. (GraphPad Prism software "line 114", Annexin V FIT-C/PI stain assay "line 122", Cell cycle analysis "line 136" and GraphPad Prism 9 software "line 148".
Reviewer 2 Report
In this work the antitumor activity of seven aryl(alkyl)tin(IV) dithiocarbamate derivatives against Jurkat E6.1, T acute lymphoblastic leukemia cells has been studied. The authors studied cytotoxic activity of tin(IV) species in relation to leukemia cells and determined IC50. The nature of cell death induced by the compounds was investigated, cell cycle analysis was carried out after treatment with the compounds, and the selectivity indexes were determined. As a result, the authors showed that six of the seven compounds have a cytotoxic effect on lymphoblastic leukemia cells. The organotin(IV) compounds, taken in IC50 concentration, induced cell death via apoptosis at 24 h of treatment. These species can induce cell cycle arrest at the different phases – G0/G1 for ODTC3 and S-G2/M for ODTC2, ODTC4, ODTC5 and ODTC7. The most promising compound ODTC3 demonstrated the lowest IC50 (0.67) and selectivity index greater than 1. This compound belongs to the triphenyl tin derivatives (known to be more toxic) and contains the lipophilic diisopropyldithiocarbamate. These results are of interest to specialists in the field of antitumor activity of compounds and this material is suitable for publication in Molecules in fields of Anticancer agents/ Metals in medicine/ Organometallics.
Small remarks:
The phrase in the ‘Discussion’ section, 2nd paragraph should be clarified:
‘The number of phenyl group in the parent structure explained the toxic effect of the organotin(IV) compounds and also characterized as electron acceptors [20].’ What is the second part of the phrase referring to? Will the Ph group of the [Sn]-Ph fragment really be a sufficiently strong electron acceptor? Or are there other types of interactions?
Page numbering are broken
After the 7th page there is figure 3 on page 8 (1?), the formatting of the picture is wrong.
Check for typos: for example, abstracts - change to "has" in the first line
